# MiR-93-5p inhibits retinal neurons apoptosis by regulating PDCD4 in acute ocular hypertension model

Cheng Tan[1,2], Wenjia Shi[1], Yun Zhang[1], Can Liu[1], Tu Hu[3,4], Dan Chen[1,3] , Jufang Huang[1,3]

**The present study focused on the effect of miR-93-5p on apoptosis of retinal neurons in acute ocular hypertension (AOH) model by regulating PDCD4 and explored its related mechanism. We detected that miR-93-5p expression was decreased and PDCD4 expression was increased in the AOH retina by qRT-PCR. Therefore, we explored the role of miR-93-5p and PDCD4. MiR-93-5p overexpression inhibited the apoptosis of retinal neurons and the expression of PDCD4 in vivo and in vitro. Inhibiting the expression of PDCD4 via transfected interfering RNA decreased the apoptosis of retinal cells and increased the expression of PI3K/Akt pathway–related proteins in vitro. However, the addition of PI3K protein inhibitor LY294002 reversed this effect, leading to a decrease of PI3K/Akt pathway protein expression and an increase of apoptosis-related protein Bax/Bcl-2 expression ratio. Finally, up-regulating miR-93-5p or down-regulating PDCD4 increased the expression of PI3K/Akt pathway protein in vivo. In conclusion, under the condition of AOH injury, miR-93-5p-inhibiting PDCD4 expression reduced the apoptosis of retinal neurons by activating PI3K/Akt pathway.**

## Introduction

Glaucoma is a neurodegenerative disease with progressive loss of retinal ganglion cells, characterized by optic nerve atrophy and defect of visual field, eventually leading to irreversible blindness (1). Glaucoma is the first irreversible blinding eye disease in the world according to the 2019 Global Disease Burden statistical report (http://www.healthdata.org/gbd/). The visual impairment and blindness caused by glaucoma increase the economic burden of society and seriously reduce the quality of life of patients (2). Elevated intraocular pressure is one of the main risk factors for glaucoma. Neurodegenerative lesions caused by pathological elevated intraocular pressure are one of the important causes of visual function decline and even blindness (3). Apoptosis is an important way of retinal neuron death in glaucoma (4). However,

the pathological mechanism of neuronal apoptosis caused by high intraocular pressure has not been fully clarified.

miRNAs are endogenous single-stranded non-coding RNAs in eukaryotes, about 22 nucleotides in length, and play a critical role in post-transcriptional regulation (5, 6). They can regulate various physiological functions, such as cell proliferation, apoptosis, development, and so on (7, 8). Previous studies have shown that the expression of miRNAs has changed significantly in glaucoma (9, 10, 11). Furthermore, more and more evidences suggested that miRNAs were important regulatory molecules of glaucomatous retinal ganglion cells apoptosis (12, 13). MiR-145-5p–down-regulating TRIM2 increased retinal ganglion cell apoptosis by inhibiting PI3K/Akt pathway in glaucoma (14). MiR-126 promoted apoptosis of rat retinal ganglion cells in a model of glaucoma through VEGF-Notch signaling pathway (15). MiR-223 induced retinal ganglion cell apoptosis and inflammatory response by reducing HSP-70 (16).

Many studies had detected miRNAs in aqueous humor, trabecular meshwork cells, and retinal tissue of glaucoma and found that the expression of many miRNAs had changed compared with the control group (10, 17, 18). We found that the expression of miR-93-5p in these three parts was significantly reduced by analyzing the types of abnormal miRNAs in these different parts, indicating that miR-93-5p may play an important role in the pathogenesis of glaucoma. MiR-93-5p is a type of miRNAs with 23 nucleotides. Numerous studies have shown that miR-93-5p can regulate neuronal apoptosis. MiR-93-5p induced apoptosis of neurons in cerebral ischemia model via inhibiting the expression of nuclear factor erythroid 2–related factor 2 (Nrf2) (19, 20). However, the regulatory effect and mechanism of miR-93-5p on apoptosis of retinal neurons are unclear in glaucoma.

Bioinformatics analysis revealed *programmed cell death protein* 4 (*PDCD4*) was a predicted target of miR-93-5p (https://cm.jefferson.edu/rna22/Interactive/, mirwalk.umm.uni-heidelberg.de/). PDCD4 was closely implicated in apoptosis (21, 22, 23). *PDCD4* is a tumor suppressor gene that regulates cell apoptosis, invasion, and tumor progression (24). MiR-340-5p alleviated neuron apoptosis during ischemia/reperfusion injury by suppressing the expression of PDCD4, whereas PDCD4 overexpression reversed the anti-apoptotic effect of miR-340-5p (25). miR-199a-3p suppressed hepatocyte apoptosis and hepatocarcinogenesis by inhibiting the expression of PDCD4 (22).

[1]Department of Human Anatomy and Neurobiology, School of Basic Medical Sciences, Central South University, Changsha, China   [2]School of Basic Medical Sciences, Hunan University of Medicine, Huaihua, China   [3]Hunan Key Laboratory of Ophthalmology, Changsha, China   [4]Department of Ophthalmology, Xiangya Hospital, Central South University, Changsha, China

Correspondence: chendan0101@csu.edu.cn

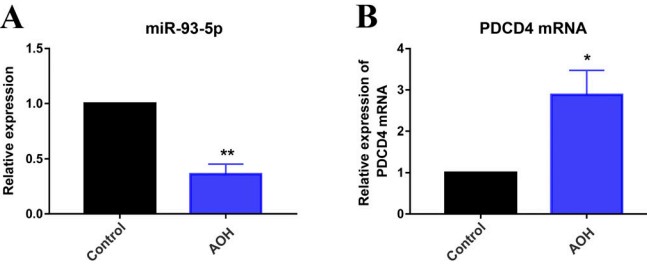

**Figure 1. Expression of miR-93-5p and PDCD4 in acute ocular hypertension (AOH) retina.**
**(A)** MiR-93-5p expression was significantly decreased compared with the control group in AOH retina (\*\*$P < 0.01$; $t$ test, n = 3). **(B)** qRT-PCR results showed that the mRNA level of PDCD4 was elevated in the AOH retina compared with the control group (\*$P < 0.05$, $t$ test, n = 3). All animals were euthanized 3 d after model establishment. Data were presented as the mean ± s.d. (n = 3). Compared with the control: \*\*$P < 0.01$, \*\*$P < 0.01$.

PDCD4 overexpression promoted the apoptosis of ovarian granulosa cells in polycystic ovary syndrome (26). MiR-93-5p can regulate PDCD4 expression in many diseases. For example, miR-93-5p promoted cells growth through inhibiting PDCD4 expression in liver cancer, gastric cancer, and nasopharyngeal carcinoma (27, 28, 29). Nevertheless, it remains unclear whether miR-93-5p can regulate the expression of PDCD4 in the acute ocular hypertension (AOH) retina.

In this study, we identified miR-93-5p as a crucial player and uncovered its specific mechanism in AOH retinal neurons apoptosis, which can potentially serve as a therapeutic target of glaucoma. Mechanistically, miR-93-5p reduced the apoptosis of retinal neurons by inhibiting PDCD4 expression. In addition, we also explored the downstream pathway of miR-93-5p/PDCD4. These findings provide new insights into the pathogenesis of glaucoma.

# Results

### Expression of miR-93-5p and PDCD4 in AOH retina

In this study, we used a rat model of AOH to explore the function and molecular mechanism of miR-93-5p, and all animals were euthanized 3 d after model establishment. The expression of miR-93-5p and PDCD4 was detected by quantitative real-time polymerase chain reaction (qRT-PCR) under the condition of AOH injury. We found that miR-93-5p expression decreased significantly ($P < 0.01$) and PDCD4 mRNA expression increased ($P < 0.05$) compared with the control group in AOH retina (Fig 1A and B). The result suggested that the expression changes of miR-93-5p and PDCD4 may play a regulatory role in retinal neuron apoptosis under the condition of AOH injury. Therefore, we preliminarily explored the effects of miR-93-5p and PDCD4 on retinal cells in vitro.

### MiR-93-5p overexpression inhibiting the expression of PDCD4 suppressed apoptosis of retinal cells through PI3K/Akt pathway in vitro

We detected the expression of miR-93-5p and PDCD4 mRNA in R28 cells after 3-h oxygen–glucose deprivation (OGD) and 12-h

reperfusion (R) using qRT-PCR. The results showed that the expression of miR-93-5p decreased ($P < 0.01$) and the expression of PDCD4 mRNA increased ($P < 0.01$) compared with the control group in OGD3 h/R12 h group (Fig 2A and B). The expression of miR-93-5p and PDCD4 mRNA was consistent with the results in vivo. Then, we examined the effect of miR-93-5p on apoptosis of R28 cells using TUNEL staining. The results of TUNEL staining showed that the apoptosis of R28 cells increased compared with the control group in OGD3 h/R12 h group ($P < 0.01$), and the apoptosis of R28 cells decreased compared with OGD3 h/R12 h group in miR-93-5p+OGD3 h/R12 h group ($P < 0.01$). However, the apoptosis of R28 cells in miR-93-5p NC+OGD3 h/R12 h group (NC meant negative control) did not change significantly compared with OGD3 h/R12 h group. TUNEL staining results demonstrated that up-regulation of miR-93-5p reduced retinal cell apoptosis (Fig 2C and D).

We further explored the mechanism of miR-93-5p regulating apoptosis in vitro. The expression of PDCD4 mRNA in R28 cells transfected with miR-93-5p was measured by qRT-PCR (Fig 2B). The results showed that the expression of PDCD4 mRNA in the miR-93-5p+OGD3 h/R12 h group decreased compared with the OGD3 h/R12 h group and the miR-93-5p NC+OGD3 h/R12 h group ($P < 0.05$, $P < 0.01$), although there was no significant difference in PDCD4 mRNA expression between the OGD3 h/R12 h group and the miR-93-5p NC+OGD3 h/R12 h group (Fig 2B). The result indicated that miR-93-5p overexpression reduced PDCD4 mRNA expression. Therefore, we further investigated the effect of PDCD4 on cell apoptosis. We transfected siPDCD4 into R28 cells and detected the expression of PDCD4 mRNA using qRT-PCR. The result showed that the expression of PDCD4 mRNA in the siPDCD4+OGD3 h/R12 h group decreased compared with the OGD3 h/R12 h group and the siRNA NC+OGD3 h/R12 h group ($P < 0.05$), although the expression of PDCD4 mRNA in the OGD3 h/R12 h group and the siRNA NC+OGD group was not statistically significant (Fig 2B). The result of qRT-PCR illustrated that transfection of siPDCD4 reduced the expression of PDCD4 mRNA in R28 cells. Then, we detected the apoptosis of R28 cells after transfection with siPDCD4 using TUNEL staining (Fig 2C and D). TUNEL-staining results showed that the number of TUNEL-positive cells in the siPDCD4+OGD3 h/R12 h group decreased compared with the OGD3 h/R12 h group and the siRNA NC+OGD3 h/R12 h group ($P < 0.05$, $P < 0.01$), although there was no significant difference between the OGD3 h/R12 h group and the siRNA NC+OGD3 h/R12 h group (Fig 2C and D). The result of TUNEL staining suggested that PDCD4 increased cell apoptosis. Taken together, miR-93-5p reduced cell apoptosis by inhibiting PDCD4 expression.

To further understand the mechanism of miR-93-5p–regulating apoptosis, we explored the pathway of miR-93-5p inhibiting R28 cell apoptosis by regulating PDCD4 expression. The levels of PI3K, Akt, p-Akt, Bax, and Bcl-2 protein in R28 cell were assessed by Western blotting (Fig 2E–H). The result showed that PI3K protein expression and p-Akt/Akt expression ratio decreased compared with the control group in OGD3 h/R12 h group ($P < 0.05$, $P < 0.01$), whereas the Bax/Bcl-2 expression ratio increased ($P < 0.01$); PI3K protein level and p-Akt/Akt expression ratio compared with OGD3 h/R12 h group in siPDCD4+OGD3 h/R12 h group increased ($P < 0.05$), whereas Bax/Bcl-2 value decreased ($P < 0.01$); PI3K protein expression and p-Akt/Akt expression ratio compared with the siPDCD4+OGD3 h/R12 h group in the siPDCD4+LY294002+OGD3 h/R12 h group decreased

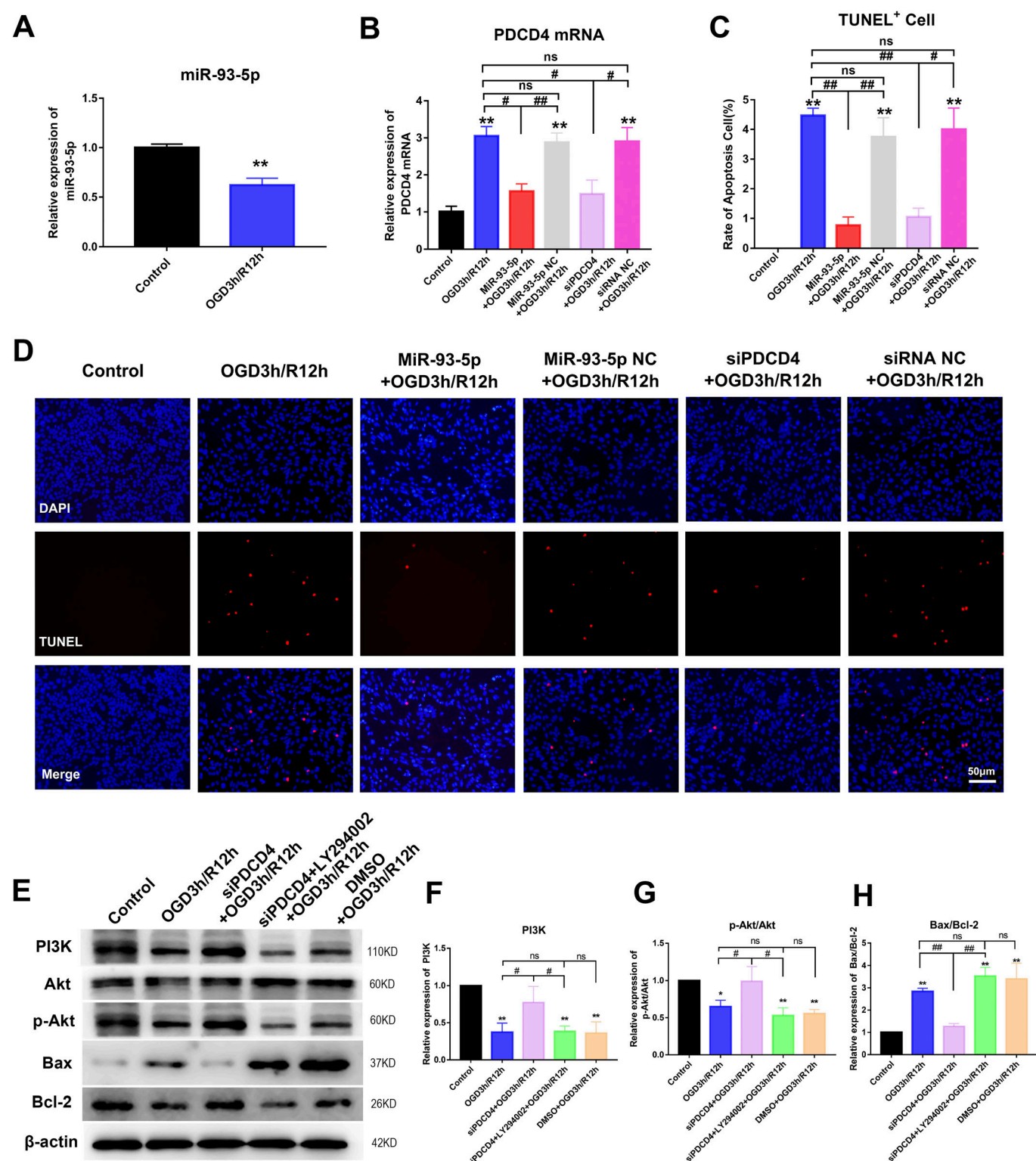

**Figure 2. MiR-93-5p overexpression inhibiting PDCD4 expression suppressed apoptosis of retinal cells through PI3K/Akt pathway in vitro.**
**(A)** The expression of miR-93-5p was detected in R28 cells after 3-h OGD and 12-h reperfusion using qRT-PCR. MiR-93-5p expression was decreased compared with the control group in OGD3 h/R12 h group (**$P < 0.01$; $t$ test; n = 3). **(B)** PDCD4 mRNA was detected by qRT-PCR (**$P < 0.01$, #$P < 0.05$, ##$P < 0.05$; one-way ANOVA; n = 3). The expression of PDCD4 mRNA increased in OGD3 h/R12 h group compared with the control group. The expression of PDCD4 mRNA was reduced in R28 cells transfected with miR-93-5p or siPDCD4 compared with the OGD3 h/R12 h group. But miR-93-5p NC or siRNA NC transfection did not have any obvious influence. **(C, D)** Apoptosis was assayed by TUNEL staining (**$P < 0.05$, #$P < 0.05$, ##$P < 0.01$; one-way ANOVA, n = 3). The findings revealed that apoptosis was significantly increased in the OGD3 h/R12 h group versus the control group. Up-regulation of miR-93-5p or down-regulation of PDCD4 expression resulted in a significant reduction in apoptosis compared with the

(*P* < 0.05), whereas the Bax/Bcl-2 expression ratio increased (*P* < 0.01). However, there was no significant difference in PI3K protein expression, p-Akt/Akt expression ratio, Bax/Bcl-2 expression ratio between the DMSO+OGD3 h/R12 h group and the OGD3 h/R12 h group. Interfering with PDCD4 expression increased PI3K protein expression and p-Akt/Akt expression ratio and decreased Bax/Bcl-2 expression ratio. However, the addition of PI3K inhibitor LY294002 (25 μM) in siPDCD4+OGD3 h/R12 h group reversed the effect of siPDCD4, resulting in a decrease of PI3K/Akt pathway proteins level and an increase of apoptosis-related proteins Bax/Bcl-2 expression ratio (Fig 2E–H). These results suggested that PDCD4 increased the apoptosis of R28 cells by inhibiting PI3K/Akt pathway. In conclusion, miR-93-5p inhibiting PDCD4 expression reduced R28 cells apoptosis through regulating the PI3K/Akt pathway.

## MiR-93-5p overexpression inhibited apoptosis of neurons in AOH retina

After we found that miR-93-5p could inhibit cell apoptosis in vitro, we did relevant research in vivo. We explored the physiological role of miR-93-5p by injecting miR-93-5p agomir and miR-93-5p agomir NC into the vitreous cavity before AOH injury. We first examined the expression of apoptosis-related molecules Bax and Bcl-2 by Western blotting. The gray scale statistical results of Bax/Bcl-2 protein bands showed that Bax/Bcl-2 expression ratio was reduced in miR-93-5p+AOH group compared with AOH group (*P* < 0.01) and miR-93-5p NC+AOH group (*P* < 0.01), and there was no significant difference in Bax/Bcl-2 expression ratio between AOH group and miR-93-5p NC+AOH group (Fig 3A and B). Then, the apoptosis of retinal ganglion cell layer (GCL) was detected by TUNEL staining. The statistical results of TUNEL-positive cells in retinal GCL showed that TUNEL-positive cells of retinal GCL were significantly decreased in miR-93-5p+AOH group compared with AOH group (*P* < 0.01) and miR-93-5p NC+AOH group (*P* < 0.01), and the number of TUNEL-positive cells of retinal GCL between AOH group and miR-93-5p NC+AOH group showed no significant difference (Fig 3C and D). Taken together, miR-93-5p inhibited neuron apoptosis in AOH retina and played a protective role in the AOH model.

## Mechanism of miR-93-5p inhibiting retinal neuron apoptosis in AOH

### Overexpression of miR-93-5p restrained PDCD4 expression

We investigated whether miR-93-5p regulated PDCD4 expression in the AOH rat model. First, the results of PDCD4 mRNA expression detected by qRT-PCR found that the expression of PDCD4 mRNA

was decreased in miR-93-5p+AOH group compared with AOH group (*P* < 0.05) and miR-93-5p NC+AOH group (*P* < 0.05), and there was no significant difference in the expression of PDCD4 mRNA between AOH group and miR-93-5p NC+AOH group (Fig 4A). Then, the statistical results of PDCD4 protein expression detected by Western blotting showed that the expression of PDCD4 protein of miR-93-5p+AOH group decreased significantly compared with AOH group (*P* < 0.01) and miR-93-5p NC + AOH group (*P* < 0.01), and PDCD4 protein expression showed no significant change between AOH group and miR-93-5p NC+AOH group (Fig 4B and C). These results demonstrated that overexpression of miR-93-5p inhibited PDCD4 expression.

### PDCD4 facilitated retinal neuron apoptosis

We explored the effect of PDCD4 on retinal neuron apoptosis in the AOH rat model. The expression of PDCD4 mRNA was detected by qRT-PCR in the AOH retina after injecting siPDCD4 into vitreous cavity. The results showed that the expression of PDCD4 mRNA decreased in siPDCD4+AOH group compared with AOH group (*P* < 0.01) and siRNA NC+AOH group (*P* < 0.01), and PDCD4 mRNA expression showed no significant change between AOH group and siRNA NC+AOH group (Fig 5A). The statistical results of the expression of PDCD4 protein detected by Western blotting showed that PDCD4 expression was reduced in siPDCD4 + AOH group compared with AOH group (*P* < 0.05) and siRNA NC+AOH group (*P* < 0.05), and PDCD4 expression showed no significant change between AOH group and siRNA NC+AOH group (Fig 5B and C). The results of qRT-PCR and Western blotting indicated that injection of siPDCD4 into vitreous cavity of AOH rats could significantly reduce the expression of PDCD4 in retina. In addition, the statistical results of Bax/Bcl-2 expression ratio detected by Western blotting showed that Bax/Bcl-2 expression ratio decreased significantly in siPDCD4+AOH group compared with AOH group (*P* < 0.05) and siRNA NC+AOH group (*P* < 0.05), and Bax/Bcl-2 expression ratio showed no significant change between AOH group and siRNA NC+AOH group (Fig 5B and D). We detected the apoptosis of AOH retinal neurons by TUNEL staining and counted the number of TUNEL-positive cells of retinal GCL. The results of TUNEL staining showed that TUNEL-positive cells of retinal GCL was reduced in siPDCD4+AOH group compared with AOH group (*P* < 0.01) and siRNA NC+AOH group (*P* < 0.01), and the number of TUNEL-positive cells of retinal GCL showed no significant difference between AOH group and siRNA NC+AOH group (Fig 5E and F). These results indicated that inhibiting PDCD4 expression reduced apoptosis of AOH retinal neurons. Taken together, miR-93-5p reduced the apoptosis of AOH retinal neurons by inhibiting the expression of PDCD4.

---

OGD3 h/R12 h group. But miR-93-5p NC or siRNA NC transfection had little effect. **(E, F, G, H)** Pathway-related proteins and apoptosis-related proteins were detected by Western blotting (*P < 0.05, **P < 0.01, #P < 0.05, ##P < 0.01; one-way ANOVA; n = 3). The PI3K protein level and p-Akt/Akt value decreased, the Bax/Bcl-2 value increased in the OGD3 h/R12 h group compared with the control group. PI3K protein level and p-Akt/Akt value increased, Bax/Bcl-2 value decreased in R28 cells transfected with siPDCD4 compared with the OGD3 h/R12 h group. After adding LY294002 (PI3K protein inhibitor, 25 μM) to R28 cells transfected with siPDCD4, the PI3K protein level, and p-Akt/Akt value decreased, Bax/Bcl-2 value increased. DMSO + OGD3 h/R12 h group (DMSO was the solvent of LY294002) had no significant difference compared with OGD3 h/R12 h group. Compared with the control: *P < 0.05, **P < 0.01. Compared with the OGD3 h/R12 h, miR-93-5p NC+OGD3 h/R12 h, siRNA NC+OGD3 h/R12 h group or siPDCD4 + LY294002 + OGD3 h/R12 h group: #P < 0.05, ##P < 0.01. n = 3 independent cultures. Data were presented as the mean ± s.d. NC, negative control; OGD3 h/R12 h, oxygen–glucose deprivation for 3 h and reperfusion for 12 h.

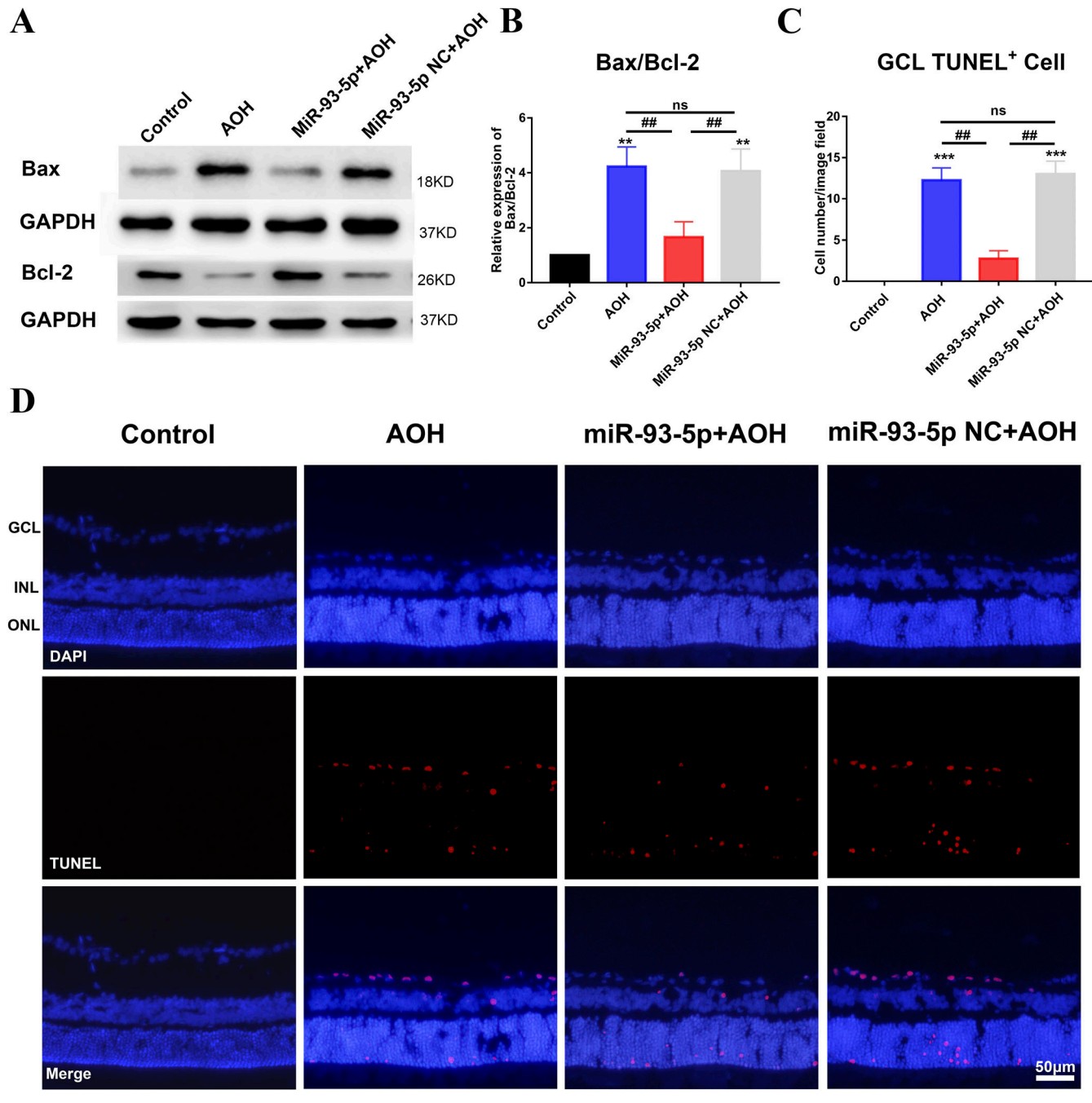

**Figure 3. MiR-93-5p suppressed neuronal apoptosis in the acute ocular hypertension (AOH) retina.**
**(A, B)** Western blot results revealed that the expression of Bax/Bcl-2 was reduced in miR-93-5p+AOH group compared with AOH group and miR-93-5p NC+AOH group. However, injection of miR-93-5p NC did not cause significant changes of retinal Bax/Bcl-2 expression compared with the AOH group (**$P < 0.01$, ##$P < 0.01$, one-way ANOVA, n = 3). **(C, D)** As demonstrated by TUNEL staining, the apoptosis of retinal neurons was notably increased in response to AOH injury. MiR-93-5p overexpression significantly reduced apoptosis, but there was no significant difference in neuronal apoptosis between miR-93-5p NC+AOH group and AOH group (***$P < 0.001$, ##$P < 0.001$, one-way ANOVA, n = 3). All animals were euthanized 3 d after model establishment. Data were presented as the mean ± s.d. (n = 3). Compared with the control: **$P < 0.01$, ***$P < 0.001$. Compared with the AOH and miR-93-5p NC+AOH group: ##$P < 0.01$. GCL, ganglion cell layer; INL, inner nuclear layer; ONL, outer nuclear layer.

### MiR-93-5p regulating PDCD4 inhibited AOH retinal neuron apoptosis, which was related to PI3K/Akt pathway

PI3K/Akt pathway had been reported to be associated with apoptosis (30) and could be regulated by miR-93-5p and PDCD4 (31, 32). Besides, our results found that PI3K/Akt pathway was the downstream mechanism of PDCD4-inducing cell apoptosis in vitro (Fig 2E–H). To further test it in vivo, we examined the expression of PI3K, p-Akt, and Akt by Western blotting in the AOH retina. The statistical results of PI3K, p-Akt, and Akt protein bands showed that the expression of PI3K and p-Akt/Akt was increased in miR-93-

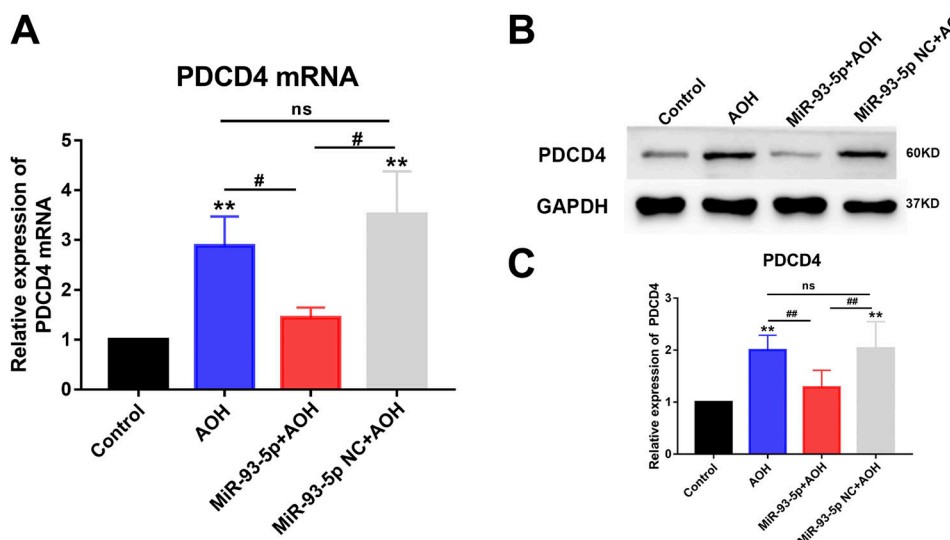

**Figure 4. MiR-93-5p negatively regulated the expression of PDCD4.**
**(A)** qRT-PCR results showed that the expression level of PDCD4 mRNA increased after AOH injury. miR-93-5p overexpression suppressed PDCD4 mRNA expression. The level of PDCD4 mRNA expression was not significantly altered in the NC+AOH group compared with the AOH group (\*\*$P < 0.01$, #$P < 0.05$, one-way ANOVA, n = 4). **(B, C)** Western blot results revealed that PDCD4 protein expression was elevated in the retina after AOH injury. Overexpression of miR-93-5p in the AOH retina significantly inhibited PDCD4 expression (\*\*$P < 0.01$, ##$P < 0.01$, one-way ANOVA, n = 3). Intravitreal injection of miR-93-5p NC did not cause significant alterations. All animals were euthanized 3 d after model establishment. Data were presented as the mean ± s.d. Compared with the control: \*$P < 0.05$, \*\*$P < 0.01$. Compared with the AOH and miR-93-5p NC+ AOH group: #$P < 0.05$, ##$P < 0.01$.

5p+AOH group compared with AOH group ($P < 0.05$) and miR-93-5p NC+AOH group ($P < 0.01$) and in siPDCD4+AOH group compared with AOH group ($P < 0.05$, $P < 0.01$) and siRNA NC+AOH group ($P < 0.05$), and there was no significant difference in the expression of PI3K and p-Akt/Akt in AOH group compared with miR-93-5p NC + AOH group and siRNA NC + AOH group (Fig 6A–F). In conclusion, the findings indicated that miR-93-5p-regulating PDCD4 expression inhibited AOH retinal neuronal apoptosis, which was related to the PI3K/Akt pathway.

## Discussion

High intraocular pressure is one of the important risk factors for glaucoma. Elevated intraocular pressure compressing the retina leads to disturbances of retinal homeostasis, for example, producing excess free radicals and excitatory amino acid and exhausting nutrient factor. The death of retinal ganglion cells is caused by these factors. However, when the high pressure was removed and the blood supply was restored, the retinal injury still continued to be worse rather than being relieved, resulting in ischemia reperfusion injury. Ischemia reperfusion injury induced by acute high intraocular pressure causes changes in the retinal microenvironment and abnormal expression of various molecules, which eventually induces apoptosis of retinal neurons. miRNAs are among the molecules with abnormal expression. They are involved in the pathological progression of glaucoma, such as miR-708 (33), miR-335-3p (33), miR-141-3p (34), miR-200a (35), etc. Our results found that the expression of miR-93 decreased in the AOH retina, which is consistent with the previously reported results (18, 36), but its regulatory effect on retinal neuron apoptosis has not been fully clarified. Previous studies have shown that miR-93-5p regulates cell apoptosis in a variety of diseases (37, 38, 39, 40). It is unclear whether miR-93-5p can directly regulate the apoptosis of retinal neurons. In the present study, we observed that miR-

93-5p inhibited apoptosis of retinal neurons in vitro and in vivo experiments. Overexpression of miR-93-5p significantly inhibited the apoptosis of AOH retinal neurons. However, previous studies found that miR-93-5p had a proapoptotic effect in cerebral ischemic injury (19, 20), which was the opposite of our results found in the AOH model. We considered that the reasons for the contrary results of previous studies to this study were as follows. First, the function of miR-93-5p was related with its regulating genes. The gene we explored had different functions from that explored by previous studies. Previous studies found that miR-93-5p increased apoptosis by inhibiting the expression of Nrf2 which had neuroprotective effects in cerebral ischemic injury. In this study, we selected a gene which was targeted by miR-93-5p and related to apoptosis and investigated its effect on apoptosis in AOH retinal neurons. Second, the type of disease in this study was different from previous studies. Previous studies explored the function of miR-93-5p in the cerebral ischemic injury model, whereas this study in AOH model.

Bioinformatics analysis indicated that *PDCD4* may be a target of miR-93-5p (https://cm.jefferson.edu/rna22/Interactive/, mirwalk.umm.uni-heidelberg.de/). Our results in the AOH rat model suggested that PDCD4 expression was inhibited by miR-93-5p. *PDCD4* is a tumor suppressor protein that inhibits translation by binding to translation initiator eIF4A, thereby promoting apoptosis and inhibiting tumor cell proliferation and invasion (21, 41, 42). PDCD4 is a regulatory factor and can increase apoptosis. PDCD4 might serve as a hub regulatory molecule that promoted neuronal apoptosis within central nervous system under neuroinflammatory conditions (43). PDCD4 facilitated lung tumor cell apoptosis by inhibiting p62-Nrf2 signaling pathway and up-regulating Keap1 expression (44). After the expression of PDCD4 was inhibited by miR-499a-5p, neuronal apoptosis decreased in ischemic brain injury (45). In this study, we found that miR-93-5p overexpression significantly reduced the expression of PDCD4, which inhibited the apoptosis of AOH retinal neurons (Figs 2–4). Our results indicated that miR-93-5p reduced PDCD4 expression and decreased the occurrence of apoptosis induced by PDCD4, thereby protecting AOH retinal neurons.

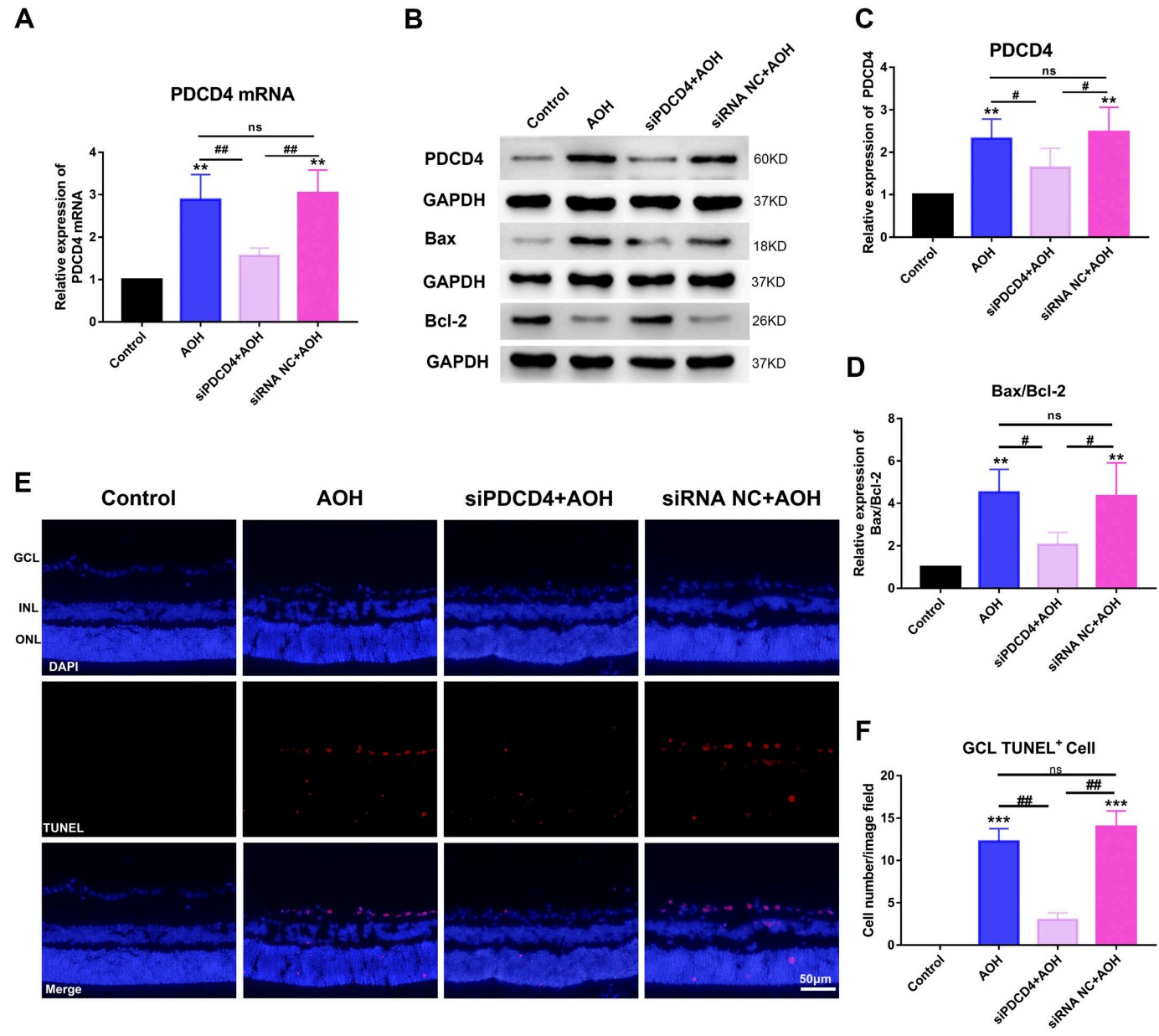

**Figure 5. PDCD4 increased neurons apoptosis in acute ocular hypertension (AOH) retina.**
**(A)** qRT-PCR was conducted to detect PDCD mRNA expression levels. PDCD4 mRNA expression was increased in the AOH group compared with the control group. PDCD4 mRNA expression levels were significantly reduced after siPDCD4 injection in AOH eyes, whereas siRNA NC injection caused no change (**$P < 0.01$, ##$P < 0.01$, one-way ANOVA, n = 4). **(B, C, D)** Western blot was performed to detect the expression of PDCD4, Bax/Bcl-2 in the retina. Compared with the control group, PDCD4 and Bax/Bcl-2 expression was increased in the AOH group. After intravitreal injection of siPDCD4, the expression of PDCD4 and Bax/Bcl-2 was decreased in the AOH retina (**$P < 0.01$, #$P < 0.05$, one-way ANOVA, n = 3). However, after injection of siRNA NC into AOH eyes, these molecules were not significantly altered compared with the AOH group. **(E, F)** TUNEL staining was used to detect the apoptosis of retinal neurons in AOH (***$P < 0.001$, ##$P < 0.01$, one-way ANOVA, n = 3). Neuronal apoptosis was significantly increased in the AOH-injured retina. After injecting siPDCD4 into the vitreous cavity of AOH, the apoptosis of retinal neurons decreased significantly. Whereas, no significant change occurred in the siRNA NC group compared with the AOH group. All animals were euthanized 3 d after model establishment. Data were presented as the mean ± s.d. Compared with the control: **$P < 0.01$, ***$P < 0.001$. Compared with the AOH and siRNA NC+ AOH group: #$P < 0.05$, ##$P < 0.01$.

The phosphatidylinositol 3-kinase (PI3K)/protein kinase B (AKT) signaling pathway plays an important regulatory role in cellular proliferation, differentiation, apoptosis, and growth (46, 47, 48, 49). Here, we focus on the role of the PI3K/Akt pathway in apoptosis regulation. Many studies have previously reported that the PI3K/Akt pathway is involved in the apoptotic regulation of glaucomatous retinal neurons. MiR-149

inhibited glaucomatous retinal ganglion cell apoptosis through activation of the PI3K/Akt pathway (50). The decreased expression of TRIM2 induced the inhibition of PI3K/Akt pathway, which contributed ganglion cell apoptosis (14). In addition, previous studies had reported that miR-93-5p or PDCD4 regulated PI3K/Akt pathway in lung cancer, glioblastoma, brain stroke, and other diseases (25, 31, 32, 51). In addition, Li et al (52)

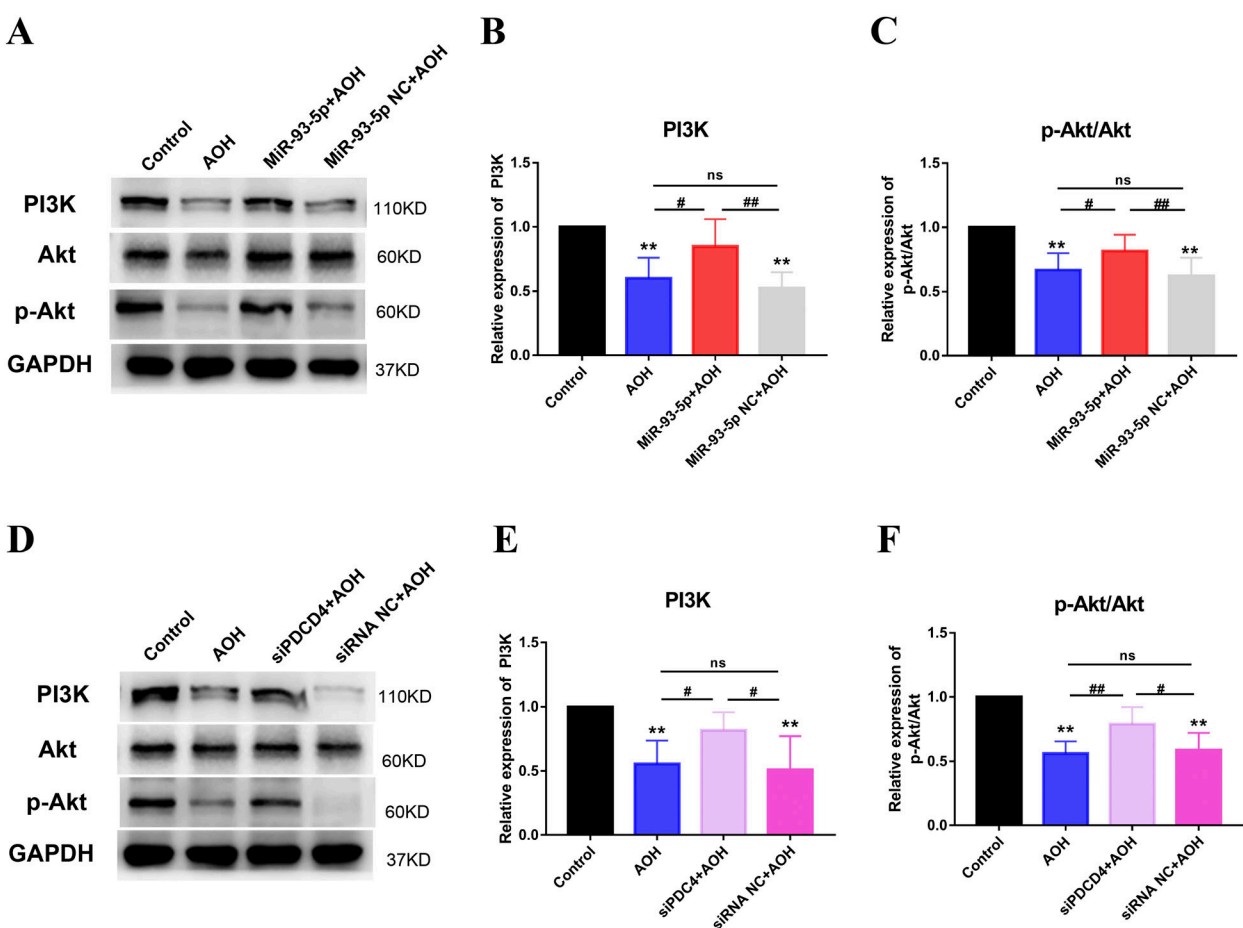

**Figure 6. MiR-93-5p regulating PDCD4 expression repressed neuronal apoptosis through activating PI3K/Akt pathway in acute ocular hypertension (AOH) retina.**
**(A, B, C)** Western blot was performed to detect the expression levels of PI3K, Akt, and p-Akt in AOH retina. Retinal PI3K, Akt, and p-Akt expression was significantly reduced in AOH injury (**$P < 0.01$, #$P < 0.05$, ##$P < 0.01$, one-way ANOVA, n = 3). Overexpression of miR-93-5p increased the expression of these proteins. These molecules showed no significant change in miR-93-5p NC+AOH group compared with AOH group. **(D, E, F)** Western blot was carried out to examine the expression levels of PI3K, Akt, and p-Akt in AOH retina. Intravitreal siPDCD4 injection of AOH increased PI3K, Akt, and p-Akt expression compared with the AOH group (**$P < 0.01$, #$P < 0.05$, ##$P < 0.01$, one-way ANOVA, n = 3). However, injection of siRNA NC did not have any obvious effect on these molecules in AOH retinal injury. All animals were euthanized 3 d after model establishment. Data were presented as the mean ± s.d. Compared with the control: **$P < 0.01$. Compared with the AOH, miR-93-5p NC+AOH and siRNA NC+AOH group: #$P < 0.05$, ##$P < 0.01$.

found that miR-93-5p expression was decreased in N-methyl-D-aspartate–treated glaucoma rats and RGCs in vitro. Overexpression of miR-93-5p, targeting phosphatase and tensin homologue, inhibited autophagy and apoptosis of RGCs by the PI3K/Akt/mTOR pathway. Our results of cell experiments demonstrated that the down-regulation of PDCD4 suppressed apoptosis of retinal neurons through the PI3K/Akt pathway (Fig 2), which was confirmed in the AOH rat model (Fig 6). The up-regulation of miR-93-5p inhibited the expression of PDCD4 in vivo and in vitro (Figs 1 and 2). Taken together, these results demonstrated that the up-regulation of miR-93-5p, which inhibited PDCD4 expression, suppressed the apoptosis of retinal neurons through the PI3K/Akt pathway. Based on our results and Li et al's findings (52), miR-93-5p can target multiple genes to regulate the PI3K/Akt pathway, which inhibited apoptosis of retinal neurons.

In conclusion, the anti-apoptotic role of miR-93-5p was demonstrated in retinal AOH injury. Mechanically, miR-93-5p suppressing PDCD4 expression reduced retinal neuronal apoptosis by activating the PI3K/Akt pathway in the rat AOH model. Our study

improved the understanding of the pathogenesis of glaucoma. Moreover, findings of this research suggested that miR-93-5p served as a potential target for glaucoma treatment. However, the present study has some limitations. First of all, there are various types of glaucoma, including elevated intraocular pressure and normal intraocular pressure types, and we were only exploring the elevated intraocular pressure type of glaucoma. Second, because our results were based on cell culture studies and experimental animal exploration, they cannot be directly applied to patients with glaucoma.

# Materials and Methods

## Establishment of the AOH model

Adult male Sprague–Dawley (SD) rats, with a mean weight of 200–250 g, obtained from the Animal Experiment Center of Central

South University, were used in the present study. We provided a comfortable living environment for the experimental animals, ensuring that they had a 12-h light/dark cycle, adequate food, and clean water. All protocols for animal use followed the guidelines for the Care and Use of Laboratory Animals issued by the National Institutes of health of the USA.

The experiment on the rat model of AOH was performed as described in the previous literature (53). SD rats were anesthetized by intraperitoneal injection of 2% pentobarbital sodium (dose: 0.2 ml/100 g/rat). Subsequently, the eyes of rats were cleaned with chloramphenicol eye drops followed by topical anesthesia with oxybuprocaine hydrochloride eye drops and dilated pupils with compound tropicamide eye drops. A 30G infusion needle was inserted into the anterior chamber of the eyes, which was connected to a saline infusion device. The intraocular pressure was slowly elevated to 110 mmHg and maintained for 60 min. Then, the elevated intraocular pressure was gradually lowered to normal pressure. Finally, the experiment on the model was completed by pulling out the needle. All animals were euthanized 3 d after model establishment.

### Intravitreous injection

A total of 60 adult male SD rats weighing 200–250 g were divided into six groups using a random number table: the control group, AOH group, miR-93-5p agomir group with AOH (miR-93-5p+AOH), miR-93-5p agomir negative control group with AOH (miR-93-5p NC+AOH), siPDCD4 group with AOH (siPDCD4+AOH), and siRNA negative control group with AOH (siRNA NC+AOH).

MiR-93-5p agomir (chemically modified miRNA agonist), siPDCD4 (chemically modified small interfering RNA sequence inhibiting PDCD4 expression), agomir NC, and siRNA NC (random sequence) were synthesized by Sangon Biotech. MiR-93-5p agomir, siPDCD4, agomir NC, and siRNA NC (0.5 nmol/$\mu$l) were injected into the vitreous cavity of the corresponding group of AOH rats 30 min before modeling. Intravitreal injection was performed according to previous literature reports (36, 54). Briefly, intravitreal injection was performed between two vortex veins about 1 mm from the corneal limbus.

### Tissue preparation

Morphological tissue was prepared as follows. First, the eyeballs of SD rats were removed after perfusion. Then, the eyeballs were postfixed for 2 h and immersed in ascending sucrose according to 15% and 30%. Finally, the eyeballs were embedded after removing the lens. The embedded eyeballs were sliced at 10 $\mu$m thickness. Then, sections were pasted on the adhesion slides, air-dried, and stored in −20°C for standby. Fresh retina was prepared. After euthanasia of SD rats, retinas were removed and placed in centrifuge tubes on ice, and the tubes were stored at −80°C for subsequent experiments.

### Cell culture and transfection

The R28 cells (rat embryonic precursor neuroretinal cells) used in this study were kindly donated by Associate Researcher Lei Shang from the Affiliated Eye Hospital of Nanchang University. The R28 cells were cultured in low-glucose DMEM containing 10% fetal bovine serum (Gibco) and incubated in a humidified incubator at 37°C with 5% $CO_2$.

R28 cells were starved for 24 h before various treatments. R28 cells at 70% confluency were used for transfection. 50 nM of miR-93-5p agomir, miR-93-5p NC, siPDCD4, and siRNA NC were transfected into cells using Lipofectamine 2000 transfection reagent (Invitrogen) following the manufacturer's manuals for 24 h.

### Transfection and oxygen–glucose deprivation/recovery (OGD/R)

R28 cells were cultured in glucose-free DMEM (Gibco), exposed under hypoxic conditions (replacing the atmosphere in the container with a gas mixture of 95% $N_2$ and 5% $CO_2$ at the speed of 3 liters/min for 5 min until the $O_2$ is rare) in a 5 liters closed container at 37°C for 3 h, and then returned to low-glucose DMEM with 10% (FBS, Gibco) in an incubator at 5% $CO_2$/37°C for 12 h. Control cells were cultured under normal conditions for the same amount of time. Three independent experiments were performed.

### Western blotting

RIPA lysis buffer (P0013B; Beyotime) containing protease inhibitor (Cwbio) and phosphatase inhibitor (Cwbio) was added to the retinal tissue. Then, the tissue was homogenized, and the supernatant was obtained by centrifugation. The protein concentration was determined by BCA protein analysis kit (Beyotime). Retinal protein (20 $\mu$g) was separated by SDS–PAGE gels, and then transferred to nitrocellulose membranes (Millipore). The membranes were blocked with 5% skim milk or BSA (4240GR100; BioFroxx) for 2 h at room temperature, then washed three times with PBST (PBS:Tween-20 = 2,000:1) and incubated with primary antibodies overnight at 4°C. The primary antibodies used in this study include anti-Bax (1:4,000, 50599-2-Ig; Proteintech), anti-Bcl-2 (1:1,000, ab59348; Abcam), anti-PDCD4 (1:1,000, 9,535; Cell Signaling Technology), anti-GAPDH (1:5,000, CM01001M; Cwbio), anti-p-Akt (1:2,000, 66444-1-Ig; Proteintech), anti-Akt (1:2,000, 60203-2-Ig; Proteintech), anti-PI3K (1:500, 20584-1-AP; Proteintech). The primary antibody on the membrane was washed off by PBST, and the membrane was incubated with rabbit and mouse HRP-conjugated secondary antibody (1:4,000, code:315-005-003, code:211-009-109; Jackson Immuno Research) at room temperature for 2 h; finally, the bands were detected by chemiluminescence technique. The relative protein expression level was measured by calculating the band gray value with ImageJ software.

### qRT-PCR

Total RNA in fresh retinal tissue was extracted with TRIzol (TSP401; Tsingke Biotechnology). Stem-loop RT primers and qRT-PCR primers (forward and reverse) of miR-93 were synthesized by Tsingke Biotechnology. Other primers included U6, GAPDH, and PDCD4. MiR-93, U6, GAPDH, and PDCD4 were reverse transcribed using RevertAid First Strand cDNA Synthesis Kit (K1622; Thermo Fisher Scientific). qRT-PCR was performed using 2×TSINGKE Master qRT-PCR Mix (TSE203; Tsingke Biotechnology). U6 and GAPDH were used as endogenous controls. The data were analyzed and

**Table 1. Primer sequences.**

| Gene | | Sequence (from 5-terminal to 3-terminal) |
|---|---|---|
| miR-93-5p | Stem-loop primers | GTCGTATCCAGTGCAGGGTCCGAGGTATTCGCACTGGATACGACCTACCT |
| | Forward | CGCAAAGTGCTGTTCGTGC |
| | Reverse | AGTGCAGGGTCCGAGGTATT |
| U6 | Forward | CTGTGGAGAAGGGAGGGTGAGAG |
| | Reverse | AGGTGAGAAGGAGGTGCAGACTG |
| PDCD4 | Forward | AACTATGATGATGACCAGGAGAAC |
| | Reverse | GCTAAGGACACTGCCAACAC |
| GAPDH | Forward | CGGCAAGTTCAACGGCACAG |
| | Reverse | GAAGACGCCAGTAGACTCCACGAC |

calculated using $2^{-\Delta\Delta CT}$ method. The primers used were as follows (Table 1).

### TUNEL staining

TUNEL staining was performed on frozen sections of eyes according to the manufacturer's protocol of the TUNEL BrightRed Apoptosis Detection Kit (A113-02; Vazyme) to detect retinal neuronal apoptosis. The staining results were analyzed, and those stained with both DAPI and TUNEL were considered as apoptotic cells, and apoptosis of retinal GCL were counted by ImageJ. One image was randomly captured from each retina for analysis, and more than three rats were set in each group.

### Statistical analysis

For in-cell experiments, we used three independently cultured cells per group. The experiments were independently repeated three times. The differences between two groups were compared using $t$ test, and comparisons among multiple groups were analyzed by one-way ANOVA, followed by Bonferroni's post hoc test. All data are presented as the mean ± SD (s.d.) and were analyzed using GraphPad Prism 7. Two-sided $P$-values less than 0.05 were considered statistically significant.

## Supplementary Information

## Acknowledgements

This research was supported by the Natural Science Foundation of Hunan Province (2021JJ30891).

### Author Contributions

C Tan: data curation, formal analysis, validation, investigation, visualization, and writing—original draft, review, and editing.

W Shi: visualization and methodology.
Y Zhang: investigation and methodology.
C Liu: investigation and methodology.
T Hu: methodology.
D Chen: conceptualization, resources, supervision, funding acquisition, methodology, project administration, and writing—review and editing.
J Huang: methodology.

### Conflict of Interest Statement

The authors declare that they have no conflict of interest.

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
