## [Reviewer comments · Life Science Alliance]

Life Science Alliance

MiR-93-5p inhibits retinal neurons apoptosis by regulating PDCD4 in acute ocular hypertension model

Cheng Tan, Wenjia Shi, Yun Zhang, Can Liu, Tu Hu, Dan Chen, and Jufang Huang

DOI: <https://doi.org/10.26508/lsa.202201732>

Corresponding author(s): *Dan Chen, Central South University*

Review Timeline:

Submission Date:	2022-09-22
Editorial Decision:	2022-12-09
Revision Received:	2023-04-27
Editorial Decision:	2023-05-26
Revision Received:	2023-05-31
Accepted:	2023-06-01

Scientific Editor: Novella Guidi

Transaction Report:

December 9, 2022

Re: Life Science Alliance manuscript #LSA-2022-01732

Prof. Dan Chen
Central South University
No. 172, Tongzipo Road, Yuelu District
Changsha 410013
China

Dear Dr. Chen,

Thank you for submitting your manuscript entitled "MiR-93-5p inhibits retinal neurons apoptosis by regulating PDCD4 in acute ocular hypertension model" to Life Science Alliance. The manuscript was assessed by an expert reviewer, whose comments are appended to this letter. We invite you to submit a revised manuscript addressing the Reviewer comments.

When submitting the revision, please include a letter addressing the reviewer's comments point by point.

Thank you for this interesting contribution to Life Science Alliance. We are looking forward to receiving your revised manuscript.

Sincerely,

B. MANUSCRIPT ORGANIZATION AND FORMATTING:

Reviewer #1 (Comments to the Authors (Required)):

Summary,

The authors present work assessing the role of miR-93-5p in the apoptotic death of retinal ganglion cells (RGCs). The group showed that miR-93-5p levels fall in an acute model of pressure-induced retinal injury. At the same time, expression of a potential miR-93-5p target, PDCD4, increase. As PDCD4 is involved in apoptosis, the group tested the hypothesis that miR-93-5p inhibits PDCD4 to regulate apoptosis through a series of functional experiments using injection of miR-93-5p or siRNA knockdown of PDCD4. Finally this mechanism is linked to the activation of PI3k/AKT signaling. Overall, the work is logical and the experiments are performed with multiple replicates, using a variety of molecular and biochemical approaches. The topic of miRNA mediated regulation of apoptosis in retinal ganglion cells is of interest. However, there are substantial issues with the text and analyses that need to be addressed. Key experimental details are missing throughout, such as the time points of analyses and the details of TUNEL quantification. Also, statistical analyses are not performed properly for multiple experimental groups. Finally, the discussion and introduction are missing important nuances and relevant literature regarding the model and pathways that all need to be addressed. Detailed questions regarding these points are described below:

Major Points:

1. Figure 1: What is the time point of these data? What do the error bars represent?
2. Figure 2: What does 'NC' mean - 'non-coding'? Based on context it seems like these are intended to be the controls for each treatment, but that is never explained in the text. Also, what do the graph error bars represent and how were the statistics analyzed?
3. Figure 2: Was there any analyses of PDCD4 expression to confirm the knockdown? In addition, it would be interesting to see if the miR-93-5p transfection alters PDCD4 levels.
4. Figure 2: Based on Figure 1, I would have thought that increase PDCD4 would increase apoptosis (increased TUNEL signal), but in vitro the effect is the opposite (PDCD4 knockdown reduced OGD apoptosis. No explanation is offered for this discrepancy.
5. Figure 3: These multiple-group data should be analyzed statistically by ANOVA, not a student's T-test as indicated. Again, what is the timepoint for these analyses?
6. Figures 4, 5, 6: What is the timepoint for these analyses? ANOVA analyses should be used for comparing multiple groups, not a student's t-test. (Minor point, note: For Figure 5C the graph uses the term "HIOP" whereas all the other graphs use "AOH")
7. Figure 6: What is the proposed connection between PI3K/AKT and miR-93-5p/PDCD4? These data only show a correlation in levels.
8. The discussion describes the acute IOP model as a mechanism that "compresses the retina and optic nerve" (p. 9, l. 184). However, this is not how most researchers view this model. Because the elevated pressure (110mmHg) is higher than the retinal vascular perfusion pressure, it causes acute ischemia-reperfusion injury in addition to any biomechanical damage. The ischemia-reperfusion injury is generally thought to drive the RGC apoptosis in this model. These results might still be relevant to glaucoma as a vasoregulatory and metabolic injury. However, the introduction cites several papers (refs 19-21) showing that miR-93-5p increased neuronal apoptosis in cerebral ischemia models. These results would seem to be the opposite of the results described here in an IOP-ischemia model (miR-93-5p inhibits apoptosis). Can the authors offer an explanation for these seemingly contrasting results?
9. In the discussion or introduction. More information about PDCD4 would be helpful - what is it, and what is known about its role in apoptotic mechanisms?
10. A paper published in 2018 (PMID: 29421576) describes miR-93-5p mediated protection of RGCs in an excitotoxic model via AKT. Relevance of this work to this prior literature should be addressed in the discussion.

11. The methods for quantification of the TUNEL signal in retinal sections needs more details. How many sections per eye were analyzed? What portion of the retina was quantified?

Minor Points:

1. The paper could use an additional round of editing throughout in order to make the language more clear.

Thank you very much for your comments and suggestions. Those comments are all valuable and very helpful for revising and improving our paper, as well as the important guiding significance to our researches. We carefully made revisions to the paper according to the suggestions. The main corrections in the paper and the responds to the reviewer's comments are as following:

Responds to the reviewer's comments:

Reviewer #1:

Major Points:

1. Figure 1: What is the time point of these data? What do the error bars represent?

Response:

Thanks for this comment. This information was detailed in the methods of this paper. All animals were sacrificed 3 days after model establishment (p. 16, l. 328). Data were presented as the mean \pm s.d. (p. 20, l. 416-418). Your suggestions are really good. According to your suggestions, this information have been added in the Figure 1 legends (p. 24, l. 574). Besides, this information has been added to other Figure legends as well.

2. Figure 2: What does 'NC' mean - 'non-coding'? Based on context it seems like these are intended to be the controls for each treatment, but that is never explained in the text. Also, what do the graph error bars represent and how were the statistics analyzed?

Response:

Thanks for your comments. The meaning of 'NC' was explained in the methods (p. 16, l. 333-334), and it meant negative control in this paper. Your comments are helpful for us. All multiple-group data have been analyzed statistically by one-way ANOVA in this study. Data were presented as the mean \pm s.d.. According your suggestions, this information has been added in the Figure 2 legends (p. 24, l. 577-578).

3. Figure 2: Was there any analyses of PDCD4 expression to confirm the knockdown? In addition, it would be interesting to see if the miR-93-5p transfection alters PDCD4 levels.

Response:

Thanks for your comments and they are very useful. we have supplemented the relevant experiments in vitro (Figure 2B). PDCD4 mRNA was detected by qRT-PCR. The expression of PDCD4 mRNA increased in the OGD3 h/R12 h group compared with the control group. And the expression of PDCD4 mRNA was reduced in R28 cells transfected with miR-93-5p or siPDCD4 compared with the OGD3 h/R12 h group. But miR-93-5p NC or siRNA NC transfection did not have any obvious influence (** $p < 0.01$; one-way ANOVA, $n=3$) (p. 24, l. 581-586). These results confirm the knockdown of PDCD4, and suggested that miR-93-5p inhibited the expression of PDCD4 mRNA.

4. Figure 2: Based on Figure 1, I would have thought that increase PDCD4 would increase apoptosis (increased TUNEL signal), but in vitro the effect is the opposite (PDCD4 knockdown reduced OGD apoptosis. No explanation is offered for this discrepancy.

Response:

Thanks for your comment. In this study, the experimental results in Figure 2 are consistent with those in Figure 1. But we didn't detect the result of PDCD4 expression in vitro. And it made the results of Figure 2 difficult to understand. Therefore, we have supplemented the experiment detecting PDCD4 expression in vitro (Figure 2B). We make the following explanation based on the supplementary experiment. The expression of PDCD4 mRNA was increased compared with the control group in AOH retina. Therefore, we preliminarily explored the connection between PDCD4 and retinal cell apoptosis in vitro. The results of vitro experiments showed that the expression of PDCD4 mRNA and retinal cell apoptosis increased in the OGD3 h/R12 h group compared with the control group. To further investigate the

function of PDCD4, we knocked out PDCD4 and used TUNEL staining to detect retinal cell apoptosis. The results showed that the apoptosis of retinal cell knocking out PDCD4 decreased compared with the OGD3 h/R12 h group. Taken together, there is no discrepancy of the experimental results between Figure 2 and Figure 1.

5. Figure 3: These multiple-group data should be analyzed statistically by ANOVA, not a student's T-test as indicated. Again, what is the timepoint for these analyses?

Response:

Thanks for this comment. The timepoint for these analyses was described in the method (p. 16, l. 328). Your suggestion is very helpful. According to your suggestion, we have made the following revisions. All multiple-group data have been analyzed statistically by one-way ANOVA in this study (p. 20, l. 416-418). All animals were sacrificed 3 days after model establishment. And this information has been added in the Figure 3 legends (p. 25, l. 605).

6. Figures 4, 5, 6: What is the timepoint for these analyses? ANOVA analyses should be used for comparing multiple groups, not a student's t-test. (Minor point, note: For Figure 5C the graph uses the term "HIOP" whereas all the other graphs use "AOH")

Response:

Thanks for this comment. The timepoint for these analyses was described in the method (p. 16, l. 328). Your suggestion is very helpful for us. All multiple-group data have been analyzed statistically by one-way ANOVA (p. 20, l. 416-418). According to your suggestion, this information has been added in Figure 4, 5, 6 legends. Besides, we have corrected the term "HIOP" to the term "AOH" in Figure 5C.

7. Figure 6: What is the proposed connection between PI3K/AKT and miR-93-5p/PDCD4? These data only show a correlation in levels.

Response:

Thanks for your comments. We consider that PI3K/AKT pathway is the downstream

mechanism of PDCD4 inducing cell apoptosis. As you mentioned, our results only show a correlation in levels, so we have added experiments: We investigated the connection between the PI3K/AKT pathway and miR-93-5p/PDCD4 in vitro. And the results confirmed that PI3K/AKT pathway was the downstream mechanism of PDCD4 inducing cell apoptosis. These revisions can be found on page 7-8, lines 132-153, and Figure 2E in this paper. Thanks again for your helpful suggestions.

8. The discussion describes the acute IOP model as a mechanism that “compresses the retina and optic nerve” (p. 9, l. 184). However, this is not how most researchers view this model. Because the elevated pressure (110mmHg) is higher than the retinal vascular perfusion pressure, it causes acute ischemia-reperfusion injury in addition to any biomechanical damage. The ischemia-reperfusion injury is generally thought to drive the RGC apoptosis in this model. These results might still be relevant to glaucoma as a vasoregulatory and metabolic injury. However, the introduction cites several papers (refs 19-21) showing that miR-93-5p increased neuronal apoptosis in cerebral ischemia models. These results would seem to be the opposite of the results described here in an IOP-ischemia model (miR-93-5p inhibits apoptosis). Can the authors offer an explanation for these seemingly contrasting results?

Response:

Thanks for your comments. We have revised the paper according to your suggestions: First of all, we have revised the description of mechanism of the acute IOP model in the discussion (p. 11, l. 234-236). It was described as “when the high pressure is removed, and the blood supply is restored, the retinal injury still continued worsens rather than being relieved, resulting in ischemia reperfusion injury” in the discussion. Then, we discussed the reasons why the results of previous studies were different from those of this study in discussion (p. 12, l. 249-260). We considered that the reasons for the contrary results of previous studies to this study were as follows. Firstly, the function of miR-93-5p was related with its regulating genes. The gene we explored was different functions from that explored by previous studies. Previous studies found that miR-93-5p increased apoptosis by inhibiting the expression of Nrf2

which had neuroprotective effects in cerebral ischemic injury. In this study, we selected a gene which was targeted by miR-93-5p and related to apoptosis, and investigated its effect on apoptosis in AOH retinal neurons. Secondly, the type of disease in this study was different from previous studies. Previous studies explored the function of miR-93-5p in cerebral ischemic injury model, while this study in AOH model. Finally, the study (PMID: 35451930) was retracted.

9. In the discussion or introduction. More information about PDCD4 would be helpful - what is it, and what is known about its role in apoptotic mechanisms?

Response:

Thanks for your comment and we have made revisions in the introduction and discussion. We supplemented the definition of PDCD4 and its role in apoptotic mechanisms in the introduction (p. 4, l. 66-72). PDCD4 is a tumor suppressor gene that regulates cell apoptosis, invasion, and tumor progression. PDCD4 promoted apoptosis of ischemia-reperfusion neurons, liver cancer cells, and ovarian granulosa cells. Besides, we made the following revisions about PDCD4 information in the discussion (p. 13, l. 264-266). PDCD4 is a tumor suppressor protein that inhibits translation by binding to translation initiator eIF4A, thereby promoting apoptosis and inhibiting tumor cell proliferation and invasion.

10. A paper published in 2018 (PMID: 29421576) describes miR-93-5p mediated protection of RGCs in an excitotoxic model via AKT. Relevance of this work to this prior literature should be addressed in the discussion.

Response:

Thank you very much for this information. We elaborated on relevance of our study to prior literature (PMID: 29421576) in the discussion (p. 14, l. 287-298). Rui Li et al. found that miR-93-5p expression was decreased in N-methyl-D-aspartate-treated glaucoma rats and RGCs in vitro. Overexpression of miR-93-5p, targeting phosphatase and tensin homologue, inhibited autophagy and apoptosis of RGCs by the PI3K/Akt/mTOR pathway. And our research found that the upregulation of miR-93-5p, which inhibited PDCD4 expression, suppressed the apoptosis of retinal

neurons through the PI3K/Akt pathway. Our results were consistent with Rui Li et al.'s findings that miR-93-5p had an anti-apoptotic effect. Based on our results and Rui Li et al.'s findings, miR-93-5p can target multiple genes to regulate the PI3K/Akt pathway, which inhibited apoptosis of retinal neurons.

11. The methods for quantification of the TUNEL signal in retinal sections needs more details. How many sections per eye were analyzed? What portion of the retina was quantified?

Response:

Thanks for your comments. In this study, one image was randomly captured from each retina for analysis, and more than 3 rats were set in each group.

Minor Points:

1. The paper could use an additional round of editing throughout in order to make the language more clear.

Response:

Thanks for this comment and we have re-edited the language of the paper throughout.

Sincerely,

Cheng Tan

Central South University

May 26, 2023

RE: Life Science Alliance Manuscript #LSA-2022-01732R

Prof. Dan Chen
Central South University
No. 172, Tongzipo Road, Yuelu District
Changsha 410013
China

Dear Dr. Chen,

Thank you for submitting your revised manuscript entitled "MiR-93-5p inhibits retinal neurons apoptosis by regulating PDCD4 in acute ocular hypertension model". We would be happy to publish your paper in Life Science Alliance pending final revisions necessary to meet our formatting guidelines.

- please address the remaining minor comments from Reviewer 1
- please upload your Table in editable .doc or excel format
- please add an Author Contributions section to your main manuscript text

A. FINAL FILES:

B. MANUSCRIPT ORGANIZATION AND FORMATTING:

Sincerely,

Reviewer #1 (Comments to the Authors (Required)):

The authors have made a good effort to make revisions to this manuscript. In particular, they have corrected the statistical analyses throughout, added new details and experiments to clarify the knockdown work, and provided additional explanations for some of their data. I can generally recommend this paper for publication, with the additional minor comments below, which would further improve readability for items otherwise explained in the methods at the end of the manuscript.

1. Since the paper apparently uses the same model and timepoints throughout, it would be helpful to include a brief description of the key points in the beginning of the results section. Important details (eg: the timepoint of tissue collection) are otherwise buried in the methods, which come at the end of the manuscript.
2. Abbreviations and acronyms (eg: "NC") should be defined upon first use - not left to the methods section at the end to explain.

Dear editor,

Thank you very much for your comments and suggestions. We carefully made revisions to the paper according to the suggestions. We have compiled and revised the manuscript based on your comments and the requirements of the Life Sciences Alliance for manuscript preparation. The main corrections in the paper and the responds to the reviewer's comments are as following:

Responds to the comments:

-please address the remaining minor comments from Reviewer 1

1. Since the paper apparently uses the same model and timepoints throughout, it would be helpful to include a brief description of the key points in the beginning of the results section. Important details (eg: the timepoint of tissue collection) are otherwise buried in the methods, which come at the end of the manuscript.

Response:

Thanks for this comment and it is helpful for us. According to your suggestion, we have added this information of model and timepoints to the beginning of the results section (p. 5, l. 86-88).

2. Abbreviations and acronyms (eg: "NC") should be defined upon first use - not left to the methods section at the end to explain.

Response:

Thanks for this comment. We have defined all abbreviations and acronyms of the first use in this paper (p. 5, l. 98 and l. 107). Thanks again for your helpful suggestion.

-please upload your Table in editable .doc or excel format

Response:

Thanks for this comment. We have uploaded the Table in editable .doc format.

-please add an Author Contributions section to your main manuscript text

Response:

Thanks for your comment. We have added an Author Contributions section to our

main manuscript text.

Sincerely,

Cheng Tan

Central South University

June 1, 2023

RE: Life Science Alliance Manuscript #LSA-2022-01732RR

Prof. Dan Chen
Central South University
No. 172, Tongzipo Road, Yuelu District
Changsha 410013
China

Dear Dr. Chen,

Thank you for submitting your Research Article entitled "MiR-93-5p inhibits retinal neurons apoptosis by regulating PDCD4 in acute ocular hypertension model". It is a pleasure to let you know that your manuscript is now accepted for publication in Life Science Alliance. Congratulations on this interesting work.

DISTRIBUTION OF MATERIALS:

Again, congratulations on a very nice paper. I hope you found the review process to be constructive and are pleased with how the manuscript was handled editorially. We look forward to future exciting submissions from your lab.

Sincerely,
